# Pursuing Precision: Receptor Tyrosine Kinase Inhibitors for Treatment of Pediatric Solid Tumors

**DOI:** 10.3390/cancers13143531

**Published:** 2021-07-14

**Authors:** Andrew J. Bellantoni, Lars M. Wagner

**Affiliations:** Division of Pediatric Hematology/Oncology, Duke University, Durham, NC 27710, USA; andrew.bellantoni@duke.edu

**Keywords:** pediatric solid tumors, targeted therapies, precision medicine, personalized medicine, receptor tyrosine kinases, tyrosine kinase inhibitors

## Abstract

**Simple Summary:**

Children with metastatic or relapsed solid tumors remain in desperate need of better treatment since conventional chemotherapy is often ineffective and can cause long-term complications. Precision oncology offers the possibility of less toxic and more beneficial treatment through the targeting of critical molecular vulnerabilities in tumors. Small molecule inhibitors of receptor tyrosine kinases have shown impressive activity in treating tumors with activating kinase fusions that drive oncogenesis and demonstrate the potential promise of precision oncology. However, in the absence of fusions or activating mutations, the activities of these agents have been more modest and are limited by intrinsic or acquired resistance and the lack of predictive biomarkers. In this manuscript, we track the development of receptor tyrosine kinase inhibitors for treating extracranial pediatric solid tumors and discuss relevant strategies to help optimize the use of these agents.

**Abstract:**

Receptor tyrosine kinases are critical for the growth and proliferation of many different cancers and therefore represent a potential vulnerability that can be therapeutically exploited with small molecule inhibitors. Over forty small molecule inhibitors are currently approved for the treatment of adult solid tumors. Their use has been more limited in pediatric solid tumors, although an increasing number of single-agent and combination studies are now being performed. These agents have been quite successful in certain clinical contexts, such as the treatment of pediatric tumors driven by kinase fusions or activating mutations. By contrast, only modest activity has been observed when inhibitors are used as single agents for solid tumors that do not have genetically defined alterations in the target genes. The absence of predictive biomarkers has limited the wider applicability of these drugs and much work remains to define the appropriate patient population and clinical situation in which receptor tyrosine kinase inhibitors are most beneficial. In this manuscript, we discuss these issues by highlighting past trials and identifying future strategies that may help add precision to the use of these agents for pediatric extracranial solid tumors.

## 1. Introduction

Current treatment for pediatric solid tumors typically combines chemotherapy with surgery and/or radiotherapy. Chemotherapy regimens used for sarcoma, neuroblastoma, and tumors of the liver or kidney primarily include conventional cytotoxic drugs, often with combinations that are decades old and broadly applied [1,2,3,4,5]. Although many children with solid tumors become long-term survivors, those with metastatic or recurrent disease continue to fare poorly, highlighting the need for more effective therapy. In addition, long-term effects of conventional chemotherapy such as infertility, heart failure, and second malignancy can threaten the quality and duration of life following cancer cure. To improve efficacy and reduce toxicity, pediatric oncologists have looked to precision oncology, a therapeutic approach that targets specific molecular features of tumors to customize treatment decisions beyond what is dictated by histologic diagnosis. 

Precision oncology has already transformed the treatment of many adult cancers, fueled by the development of genetic testing that allows for detection of actionable molecular changes present in diseases such as breast cancer, melanoma, and lung cancer. The increasing use of genetic testing platforms has improved our understanding of the specific molecular signatures of pediatric solid tumors, allowed for the comparison between initial and relapsed specimens, and directed the development of new inhibitors for emerging targets [6,7,8,9]. In addition, genetic testing of tumor tissue has resulted in the sub-classification of some tumor types into separate strata with molecular changes and corresponding specific therapies [7,10,11]. Precision oncology employs a wide array of treatment strategies, including monoclonal antibodies as well as small molecules targeting receptor tyrosine kinases (RTKs) or intracellular proteins. RTK inhibitors comprise the largest category of targeted agents and their application for the treatment of extracranial pediatric solid tumors is the focus of this review.

RTKs are protein complexes comprised of a transmembrane component that links the extracellular receptor with the intracellular catalytic kinase domain [12]. When a ligand binds to the extracellular receptor, it induces a conformational change that promotes autophosphorylation of the intracellular kinase and subsequent activation of downstream pathways. These pathways are involved in important cellular and biologic processes such as cell proliferation and angiogenesis [12,13,14,15]. Examples of RTK families include vascular endothelial growth factor receptor (VEGFR), epidermal growth factor receptor (EGFR), fibroblast growth factor receptor (FGFR), rearranged during transfection (RET), anaplastic lymphoma kinase (ALK), hepatocyte growth factor receptor (c-MET), and the tropomyosin receptor kinase (TRK). Alterations in RTKs may promote oncogenesis and metastasis through the overexpression of functionally normal RTKs, mutations within RTKs that render them constitutively active, or chromosomal translocations resulting in fusion proteins with conformational changes [13,16,17]. 

Small molecule RTK inhibitors were some of the first examples of successful targeting of tumor-specific genetic changes, as demonstrated by the use of imatinib to inhibit the activity of the BCR-ABL fusion tyrosine kinase that defines chronic myeloid leukemia [18]. Drugs such as imatinib typically act on the intracellular domain of the RTK by interfering with the ATP binding site and preventing phosphorylation. However, since different RTKs may have similar intracellular domains, single inhibitors sometimes lack fidelity and alter the function of more than one RTK. For example, imatinib also has activity against tumors with activating mutations of *KIT*, such as gastrointestinal stromal tumors [19]. Since imatinib first received regulatory approval for use in 2001, there have been over 40 RTK inhibitors approved worldwide [16]. More detailed information on the specific activities and different mechanistic classes of RTKs can be found in other reviews [12,20].

Despite the availability of dozens of agents, there have only been a few successful adaptations to date of RTK inhibitors to treat extracranial pediatric solid tumors. This is likely because childhood solid tumors have fewer potentially targetable mutations in RTK pathways [8] and there are fewer pediatric cancer patients to participate in clinical trials. The greatest clinical benefit has been observed when treating tumors in which RTKs are activated through mutation or translocation. Outside of this context, the activity of these agents is limited by innate and acquired resistance and suffers from the lack of predictive biomarkers [16].

In this manuscript, we review seminal trials of RTK inhibitors for treating extracranial pediatric solid tumors, highlight the few established predictive biomarkers that guide therapy, and discuss the considerable knowledge gaps that remain. In addition to identifying the “who” (which patients may benefit the most from RTK inhibition), we must also discover more about the “when” (timing of therapy within the disease course) and the “how” (role for single-agent versus combination therapy) of RTK inhibition in order to optimize the precision of this treatment approach. 

## 2. Key Clinical Trials of RTK Inhibitors to Treat Pediatric Solid Tumors 

RTKs are attractive targets for inhibition because they drive oncogenesis and angiogenesis in many pediatric solid tumors through effects on tumor cells and the microenvironment [21,22,23]. The antitumor benefit of RTK inhibitors is directly related to how essential the inhibited RTK is to tumor growth and viability and how effectively the target is inhibited. The former is dependent on tumor biology and the presence of resistance mechanisms, while the latter is dependent on drug choice and pharmacokinetic considerations. These variables account for the broad range of activity seen in pediatric studies to date. A summary of selected completed studies is presented in Table 1.

### 2.1. Targeting Oncogenesis in Tumors with Mutations, Fusions, or Amplifications

The most dramatic benefits from RTK inhibition are observed when there is substantial disruption of a key signaling pathway that uniquely drives tumor growth. An example of this has been the use of larotrectinib to treat tumors driven by fusions involving one of three isoforms of neurotrophic tropomyosin receptor kinase (*NTRK*). These genes encode the tropomyosin receptor kinase proteins TRKA, TRKB, and TRKC and partner with a variety of other genes to produce fusion proteins causing constitutive activation of the TRK proteins and subsequent signaling through various downstream pathways such as Ras-Raf-MAPK and PI3K-AKT-mTOR to drive oncogenic growth. 

Larotrectinib is a highly selective agent that blocks the ATP-binding site of TRKA, B, and C receptors with a half maximal inhibitory concentration (IC_50_) of 5–11 nanomolar [52]. Preclinical testing confirms in vitro induction of apoptosis and G1 cell cycle arrest and dose-dependent tumor inhibition is observed in mice bearing tumors with *NTRK* fusions [53]. In 2018, Drilon and colleagues published a landmark study of 55 patients with *NTRK* fusions ranging in age from 4 months to 76 years and spanning 17 different tumor histologies. Remarkably, the centrally-confirmed response rate was 75%, with 71% of responses ongoing at one year [24]. The median time for response was at the first 8-week scheduled assessment and responses were independent of the specific partners involved in the translocation. These results resulted in larotrectinib approval by the US Food and Drug Administration (FDA) as a tissue agnostic therapy for patients of any age with a documented *NTRK* fusion.

*NTRK* gene fusions occur in approximately 1% of all solid tumors [54] and are spread across many tumor types in adults. In children, *NTRK* fusions are enriched in specific histologies. For example, infantile fibrosarcoma, congenital mesoblastic nephroma, and secretory carcinoma of the breast or salivary gland frequently have *NTRK* fusions identified [55,56,57]. These fusions are also present in some spindle cell sarcomas, spitzoid melanocytic tumors, and pediatric thyroid cancer [58,59]. Each of these diagnoses is rare when individually considered, however, the marked benefit seen to date with larotrectinib suggests that pediatric oncologists should consider testing for *NTRK* fusions when caring for patients with these tumor types. For infantile fibrosarcoma patients with *NTRK* fusions, the neoadjuvant use of larotrectinib may facilitate surgical resection and potentially spare young children the long-term toxicities of alkylator and anthracycline therapy [60]. This shift in treatment paradigm shows the potential impact of RTK inhibition when provided in the optimum clinical context of a highly specific agent treating an oncogene-addicted tumor. However, it should be emphasized that not all *NTRK* fusions identified by sequencing are functional or would be expected to respond to a RTK inhibitor. For example, in a genetic screen of 113 banked osteosarcoma tumor samples, three (2.7%) had *NTRK* fusions although none were functional [61]. The authors hypothesized that the inherent chromosomal instability of osteosarcoma increases the likelihood of random passenger mutations and caution should be applied when assessing the potential treatment implications of agents targeting such fusions in which there are no signs of RNA or protein expression. In fact, immunostaining using a pan-TRK antibody has emerged as a highly sensitive and specific method of screening tumor samples for possible *NTRK* fusions and some authors suggest performing immunohistochemistry first followed by molecular confirmation of positive cases as a method to better identify functional fusions [55].

Hong et al. recently reported an expansion of the initial larotrectinib study to include 153 evaluable patients aged from 1 month to 84 years [25]. Similar findings were again observed with a centrally reviewed response rate of 79%, including 16% with complete responses. The median duration of response was 35 months and responses were observed in patients with intracranial metastases, suggesting adequate central nervous system penetration. Despite the impressive response rates, resistance can still occur. Mutations in the ATP-binding site side of the kinase account for some cases of primary or acquired resistance and are termed “solvent front” mutations as they affect the solvent-exposed portion of the kinase domain and sterically interfere with drug binding [24]. These findings have resulted in the development of next-generation agents, such as selitrectinib and repotrectinib, that are specifically designed to overcome these issues [52,62,63].

Other notable successes in pediatric solid tumors include targeting the *ALK* fusions observed in the majority of anaplastic large cell lymphoma (ALCL) and approximately half of the patients with inflammatory myofibroblastic tumors (IMT) [64,65]. Crizotinib is a first-in-class ALK inhibitor that also has activity against MET and ROS1. A pediatric phase I study of crizotinib identified responses in ALCL and IMT [26], resulting in a phase II expansion of these cohorts. That study included 26 patients with relapsed/refractory ALCL and 14 patients with inoperable/metastatic IMT [27]. Responses were observed in over 80% of both cohorts and were often durable. The median time to response was within the first month of therapy. These findings resulted in the 2021 FDA approval of crizotinib for the treatment of relapsed ALCL in pediatric patients.

Crizotinib has also shown activity in some pediatric patients with *ALK* point mutations or amplification (>10 copies), which are observed in 14% of neuroblastoma patients and associated with the worse outcomes [10,66]. In this disease, *ALK* mutations occur at different sites, which correlate with variable sensitivity to crizotinib based on different ATP-binding affinities of the *ALK* mutations [67]. This laboratory finding was reflected clinically in a phase I trial of crizotinib showing response in only a subset of relapsed neuroblastoma patients [26]. One method to address resistance is to combine crizotinib with standard chemotherapy drugs and in preclinical studies this strategy produces responses even in crizotinib-resistant models [68]. This strategy is being pursued in an ongoing phase III trial for newly-diagnosed high-risk neuroblastoma patients with amplification or mutation of *ALK* who will receive crizotinib in combination with all planned adjuvant therapy (ClinicalTrials.gov Identifier: NCT03126916, Table 2).

Other strategies to address resistance include the development of second-generation and third-generation ALK inhibitors such as alectinib, ceritinib, lorlatinib, and ensartinib. Preclinical testing of these drugs has demonstrated activity in cell lines or xenograft models that harbor *ALK* mutations resistant to crizotinib [69,70,71,72] and anecdotal reports and early phase I trials showing clinical activity are now starting to appear [70,73,74,75]. Another drug, entrectinib, is a multi-targeted RTK inhibitor with action not only against ALK but also the TRK-B receptor, which is expressed in over half of high-risk neuroblastoma tumors and associated with a poor prognosis [76]. Entrectinib received FDA approval based on high response rates seen in patients 12 years and older with *NTRK*, *ROS1*, or *ALK* fusions [77]. Its impact against neuroblastoma through targeting *ALK* mutations or TRK-B expression is under further investigation [78]. More comprehensive reviews of ALK-targeted therapy in neuroblastoma are available [10,30]. The evolution of agents and ability for more precise selection based on specific mutations is similar to the development of therapies for gastrointestinal stromal tumor in which resistance to the primary RTK inhibitor (imatinib) resulted in the selective incorporation of later-generation drugs with activity against specific mutations [19].

Other RTK proto-oncogenes such as *RET* are altered in pediatric solid tumors and may be potentially targetable. *RET* mutations and fusions have been documented in pediatric thyroid cancer and are associated with a more aggressive phenotype via downstream signaling through MAPK, PI3K, and JAK-STAT [79,80]. Multi-kinase inhibitors of RET, including cabozantinib and vandetanib, have modestly improved survival in patients with medullary thyroid cancer (MTC), but their toxicity may limit the durability of benefit [81,82]. In contrast, LOXO-292 (selpercatinib) is a highly selective RET inhibitor with compelling activity even in patients previously treated with other RTK inhibitors. Furthermore, the narrow spectrum of inhibition results in less toxicity, with only 2% of patients withdrawing because of the side effects [33]. These results have resulted in approval for patients over 12 years of age with *RET*-mutant or fusion-positive advanced thyroid cancer requiring systemic therapy.

Fibroblast growth factor receptor (FGFR) is a family of RTKs with downstream targets that promote cell proliferation, survival, and migration, including MAPK and PI3K [83]. FGFR is often overexpressed or mutated in rhabdomyosarcoma [84]. Erdafitinib is a pan-FGFR inhibitor with activity at levels as low as 1 nanomolar [85] and is approved for the treatment of metastatic urothelial cancer in adults. A prospective phase II trial for pediatric solid tumors with FGFR alterations is now being conducted (NCT03210714, Table 2). FGFR signaling also plays a prominent role in angiogenesis, which provides an alternative avenue for the use of RTK inhibitors [14].

### 2.2. Targeting Angiogenesis and Non-Mutated RTKs

Even when an activating mutation or gene fusion is not identified in tumor cells, RTK inhibitors still may be useful although durable benefit is less likely. This strategy has primarily focused on targeting angiogenesis, which is the main mechanism for the new blood vessel development that is essential for tumor growth and metastasis. Although this process is complex and driven by many factors, signaling through the vascular endothelial growth factor receptor 2 (VEGFR2) is particularly important [86]. VEGFR2 is often overexpressed by the endothelial cells of solid tumors and can be inhibited at low nanomolar levels by several of the RTK inhibitors used to treat pediatric solid tumors even in the absence of a defined gene mutation or fusion. In this manner, the RTK inhibitor is acting to disrupt the supply of nutrients to the tumor rather than directly inhibiting a specific oncologic driver.

Among extracranial pediatric solid tumors, RTK inhibitors targeting angiogenesis have most often been used for therapy of recurrent sarcoma. This strategy is partly modeled after the use of pazopanib to treat adult soft tissue sarcoma. Pazopanib was initially identified through a screening process for VEGFR2 inhibitors. However, similar to many of the agents used for sarcoma, pazopanib is a multi-kinase inhibitor that also inhibits platelet-derived growth factor receptor (PDGFR) and c-KIT at <100 nanomolar, with slightly less inhibition of FGFR [87]. These additional tyrosine kinases may promote growth through mechanisms other than angiogenesis and preclinical studies in a variety of models suggest that pazopanib has activity not only in tumor-associated endothelial cells and angiogenesis but also directly on tumor cells as well as the tumor microenvironment [88]. Early phase clinical trials of pazopanib showed disease stability in adult sarcoma patients and in a randomized phase III trial, pazopanib demonstrated improved progression-free survival from 1.6 to 4.6 months when compared to the placebo in adults with advanced soft tissue sarcoma [89]. Although this finding resulted in regulatory approval, the objective response rate was low (6%) and the overall survival was not statistically different. 

The approval of pazopanib in adults stimulated further interest in using multi-RTK inhibitors for pediatric sarcomas. Supporting this was the potential importance of angiogenesis in osteosarcoma and Ewing sarcoma [90,91], as well as the expression of targetable RTKs such as RET, MET, PDGFR, KIT, AXL, and FGFR. The experience of several recent phase II trials of multi-targeted RTK inhibitors for the treatment of recurrent osteosarcoma and Ewing sarcoma has been summarized [92]. The drugs studied included sorafenib, apatinib, lenvatinib, cabozantinib, and regorafenib [35,37,38,40,42,44]. These agents all have variable inhibitory effects on the kinases noted above and several important observations can be made. First, as observed with pazopanib, objective responses were generally modest in number and nearly always partial, with the primary clinical benefit being improved progression-free survival compared to either placebo or historical controls. This higher rate of primary and secondary resistance is in contrast to studies targeting specific RTK fusions mentioned previously, in which responses were more frequent and durable. Secondly, the toxicity of these multi-RTK inhibitors can be considerable, with much higher rates of dose reductions and treatment discontinuation than observed in agents such as larotrectinib. These side effects can negatively affect quality of life even in patients who are having an oncologic response [47,93] and some toxicities such as pneumothorax seen in patients with pulmonary metastases may require hospitalization and/or invasive interventions. Third, there is variability with respect to clinical benefits between agents, suggesting that some targets may perhaps be better suited for therapy of these two diseases. Tian et al. suggested that a careful review of the level of inhibition provided for each target coupled with the results of recent clinical trials could help identify the most relevant RTKs for targeting [46]. For example, the activity of imatinib is quite limited in osteosarcoma [94], suggesting that the inhibition of KIT and PDGFR may be inferior targets. By contrast, all five of the agents with activity against bone sarcoma inhibited VEGFR2 and RET at concentrations <15 nanomolar, while imatinib’s inhibition of these targets is much more limited. This strategy of exclusion based on the spectrum of target inhibition and clinical trial performance is one method to begin the rational selection of agents, but in the absence of a specific mutation or fusion the method involves inference based on small studies of heavily pretreated patients without direct comparisons between such agents. Stated in another manner, it remains very difficult to know the critical nature of any particular target in these tumor types and it is clear that the simple expression of a target is not sufficient to guarantee a high likelihood of clinical benefit. Similarly unknown is the extent of target inhibition necessary for successful treatment and this may in part depend on pharmacokinetic and pharmacodynamic properties of the drug as well as patient tolerance. 

Despite these uncertainties, the disease stabilization observed in multiple relapsed patients does suggest that multi-RTK inhibitors may ultimately have a further role in therapy for bone sarcoma. In other pediatric solid tumors, results have been less consistent. Occasional remarkable responses have been observed in patients with recurrent neuroblastoma treated with imatinib [95], as well as multiple recurrent Wilms tumor patients with overexpression of RET and MET who responded to cabozantinib [96]. However, in a larger cooperative group phase II trial of imatinib, no responses were observed in patients with neuroblastoma, desmoplastic small round cell tumor, or synovial sarcoma [94]. Similarly, no responses were observed for sorafenib in patients with relapsed Wilms tumor or rhabdomyosarcoma [97]. In a preliminary report of a Children’s Oncology Group (COG) study of cabozantinib, activity was confirmed for patients with recurrent osteosarcoma, but not seen in those with Ewing sarcoma, rhabdomyosarcoma, other soft tissue sarcoma, or Wilms tumor [36]. Results from a COG study of pazopanib for patients with relapsed osteosarcoma, Ewing sarcoma, rhabdomyosarcoma, other soft tissue sarcoma, neuroblastoma, or hepatoblastoma have not yet been reported (NCT01956669). Finally, given that sorafenib has received regulatory approval for advanced hepatocellular carcinoma, anecdotal use of this agent has been reported in children with liver cancers [98,99]. The combination of sorafenib with conventional chemotherapy for treatment of pediatric hepatocellular carcinoma is being prospectively assessed in an ongoing international trial (NCT03017326, Table 2). 

The use of RTK inhibitors has also been investigated for treatment of other uncommon pediatric solid tumors. Although germ cell tumors have been reported to express c-KIT and PDGFR, inhibitors such as sunitinib, pazopanib, and sorafenib have shown only limited activity [100]. In contrast, sorafenib appeared beneficial for treatment of adults with desmoid tumors, with a response rate of 33% and a doubling of the 2 year PFS compared to the placebo [101]. The use of sorafenib or pazopanib is now being explored for younger patients as well [102,103].

Another method for targeting angiogenesis in solid tumors is the use of bevacizumab, a monoclonal antibody that binds to VEGF and interferes with binding to VEGFRs. This strategy has a different mechanism of inhibition that is narrower than observed with RTK inhibitors. Previous studies have shown limited activity of bevacizumab as a single agent [104] and in combination with chemotherapy for pediatric sarcoma [105,106]. However, no direct comparisons of anti-VEGF antibodies versus RTK inhibitors in pediatric solid tumor patients have been reported.

### 2.3. Targeting Tumor Microenvironment

Aberrant angiogenesis can promote a hostile tumor microenvironment that is hypoxic and acidotic [107]. RTKs have been shown to modulate non-endothelial components of the tumor microenvironment through complex mechanisms such as the enhancement of antitumor immunity and local immune cell populations [23]. One example is the role of VEGF in creating an immunosuppressive environment through the recruitment of myeloid-derived suppressor cells and regulatory T cells, as well as the inhibition of dendritic cell maturation and antigen presentation [108]. A deeper understanding of the tumor microenvironment and its role in cancer is required to aid in the design of future clinical trials. 

To begin to address these complicated changes, Wilky et al. combined the VEGFR inhibitor axitinib with the anti-programmed death 1 (PD-1) antibody pembrolizumab to treat adults with sarcoma [109]. The 3 month PFS of 65% and response rate of 25% exceeded historical expectations. Similarly encouraging preliminary results were also observed with a trial of sunitinib and the anti-PD-1 antibody nivolumab for adult bone sarcoma [110]. Additional studies are underway to further assess the ability of RTK inhibitors to synergize with immunotherapies, such as a trial of regorafenib and nivolumab for patients with recurrent osteosarcoma (NCT04803877, Table 2). However, there is a potential that combination therapy may exacerbate toxicity, as observed in a study combining crizotinib and nivolumab in lung cancer patients which was terminated early due to severe hepatic toxicity [111].

## 3. Intrinsic and Acquired Resistance

Resistance remains the greatest barrier to improving the clinical benefit of RTK inhibitors. Resistance can be intrinsic, in which tumors continue to grow despite treatment, or acquired, in which initially responding or stable tumors developed progression over time. Although pediatric tumors with mutation-driven or fusion-driven oncogenesis are often initially responsive to RTK inhibition, certain specific mutations seem inherently resistant. Examples include mutations in *ALK* (e.g., F1174L) or *NTRK* that have driven development of later-generation inhibitors described earlier. Another cause of primary resistance is the absence of protein expression despite molecular evidence of gene fusion, as observed in some patients with certain *NTRK* fusions whose tumors are resistant to larotrectinib [24]. Finally, RTK inhibitors are not effective if the target is not essential to the growth of the tumor. This issue likely is the cause for most intrinsic resistance to the multi-RTK inhibitors used for sarcoma, in which the inhibition of targets that are expressed or even amplified is not enough to stop the growth of the tumor.

Acquired resistance may also develop from several different mechanisms, such as the later development of mutations in the primary target, the development of alternate signaling pathways that drive tumor growth, or the expression of drug resistance proteins. These various scenarios are reviewed in more detail elsewhere [17,112]. Understanding these mechanisms can sometimes result in improved treatment. For example, the multi-kinase inhibitor sorafenib produced objective responses in 14% of patients with recurrent osteosarcoma, but by 6 months only 29% of patients were free of progression [44]. Upon further investigation, Pignochino and colleagues had demonstrated that sorafenib increases downstream mTORC2 signaling in osteosarcoma cells and in xenograft models this escape mechanism was abrogated with the mTOR inhibitor everolimus [113]. This laboratory observation was then translated to a clinical trial combining sorafenib and everolimus in which the 6 month progression-free survival was improved to 45% [114]. In analogous works in high-risk neuroblastoma, molecularly selected patients are treated in an ongoing study with the ALK inhibitor ceritinib combined with the CDK4/6 inhibitor ribociclib based on preclinical evidence of synergy (NCT02780128, Table 2) [115]. 

Another proposed therapeutic strategy is the creation of a single drug to target two distinct pathways that are activated together in tumor cells. In high-risk neuroblastoma, *ALK* mutations co-segregate with *MYCN* amplification, with the *ALK* mutation driving its transcription and expression. Since inhibiting both oncogenic pathways with a single drug would be desirable, investigators have begun to identify candidate dual inhibitors with preclinical activity [116]. These approaches are laudable and are the logical next steps in drug development. However, resistance mechanisms are often complex and incompletely understood and even when downstream compensatory pathways are carefully documented in preclinical models of pediatric sarcoma [117], combined inhibition of seemingly relevant pathways are not always effective [118].

In an effort to overcome intrinsic resistance and to prevent the development of acquired resistance, investigators have also combined chemotherapy with RTK inhibitors in the laboratory and in the clinic [51,68,119]. Examples include the incorporation of crizotinib along with chemotherapy for patients with newly diagnosed *ALK*-altered high-risk neuroblastoma (NCT03126916, Table 2) and the randomized phase II screening trial of ifosfamide/doxorubicin with or without pazopanib in newly diagnosed non-rhabdomyosarcoma soft tissue sarcoma [51]. In the latter study, the addition of pazopanib was feasible and resulted in improved rates of pathologic necrosis at definitive resection, although the final impact on survival has not yet been reported. Combination therapy did increase the incidence of toxicity, but this was manageable. 

Russo and colleagues combined pazopanib with a different standard chemotherapy backbone (vincristine plus irinotecan) in a single-arm study, identifying an objective response rate of 47% and median time to progression of 10 months in pediatric patients with various types of recurrent sarcoma [120]. However, when combining this same chemotherapy backbone with regorafenib, only sequential but not concomitant treatment was tolerated, perhaps because regorafenib can inhibit expression of UGT1A1 and increase toxicity of irinotecan [121]. Similarly, when pazopanib was added to temozolomide plus irinotecan, toxicity was considerable and a maximum tolerated dose could not be identified [119]. These results differed from those reported by Gaspar et al. regarding a phase I study of lenvatinib combined with ifosfamide and etoposide for patients with recurrent osteosarcoma, in which toxicity was more manageable and two-thirds of patients were progression-free at 4 months [42]. Finally, the pharmacokinetic properties of RTK inhibitors can be exploited therapeutically as shown by Furman et al. who used the EGFR inhibitor gefitinib to inhibit the drug resistance protein ABCG2, resulting in a four-fold increase in the bioavailability of oral irinotecan [122]. These early studies demonstrate that while adding RTK inhibitors to conventional chemotherapy is a rational strategy to try to overcome resistance, close attention to toxicity, dosing, sequencing, and drug interactions is necessary to optimize therapy. In addition, since these chemotherapy backbones already have some baseline level of activity, assessing the true benefit will require larger controlled studies, which are being planned by cooperative groups such as the COG for bone sarcomas. Examples of ongoing studies of RTK inhibitors to treat extracranial solid tumors are provided in Table 2.

## 4. Toxicity, Dosing, and Pharmacokinetic Considerations

The side effect profile of RTK inhibitors is different than that of conventional cytotoxic therapy and is generally related to the specific RTKs that are affected. For example, agents targeting VEGFR are commonly associated with hypertension, diarrhea, proteinuria, poor wound healing, and thyroid dysfunction, while those targeting ALK or MET may cause vomiting or pancreatic inflammation. The scope of toxicities can affect a wide range of organ systems and can include weight gain, dizziness, and pain upon withdrawal of therapy [123]. The incidence of toxicities may be related to the extent of various kinases inhibited. For example, highly-specific inhibitors such as larotrectinib and selpercatinib require dose reductions in less than 10% of patients [25,33], while one-third or more of patients receiving multi-RTK inhibitors require treatment modifications [92]. 

The degree to which on-target toxicities such as these are associated with benefits in pediatric patients is unclear [93,124,125,126] since some side effects take time to develop and are therefore observed more in patients who do not have intrinsic resistance and early progression of disease. Given the continuous administration schedule used with most RTK inhibitors, the nature of toxicities can affect patients in a different manner than observed with conventional chemotherapy in which side effects are more acute but often substantially improve within a treatment cycle. In fact, sometimes toxicity results in an overall worse quality of life despite relatively high response rates, as observed in a recent study of apatinib for pediatric patients with recurrent osteosarcoma [47]. In addition, the true prevalence of side effects associated with RTK inhibitors may be underestimated [127]. Of special concern in pediatrics is the potential for impairment of critical developmental pathways in growing children, particularly if more lengthy therapy is given [128]. Long-term monitoring of linear growth, endocrine function, and cardiac function will be important to better understand the potential late effects of these agents in young patients.

Toxicities from RTK inhibitors are used in early phase clinical trials to determine dosing, which for the majority of oncology drugs is based on a maximum tolerated dose (MTD) assessed by first-cycle toxicity in early clinical trials. Given that cumulative toxicity may be a larger issue with RTK inhibitors and given that demonstration of a steep dose–response curve may be less clear with these drugs, some investigators have advocated for establishing an optimal biological dose (OBD) that identifies the lowest dose at which the desired biological effect is observed [129]. Understanding the activity of RTK inhibitors at a range of dosing may be important when combining these agents with chemotherapy backbones since lower doses are often necessary to maintain tolerability [27,51,130]. In other settings, the dose of RTK inhibitors may be increased until some secondary side effect occurs. For example, dose-dependent elevations in serum phosphate are seen as a marker of FGFR inhibition and therefore erdafitinib activity [131]. In clinical trials, the dose of erdafitinib has been increased if patients had no treatment-related effects and failed to reach a target serum phosphate level of 5.5 mg/dL [132]. There are several important pharmacokinetic considerations related to the use of RTK inhibitors. Many of these agents are metabolized through the CYP3A4 pathway, which may vary between individuals and be affected by concurrent medications. As with any oral drug, absorption may potentially be affected by food or the use of antacids that affects gastric pH [133,134,135]. In addition, treatment of young children often requires oral solutions instead of capsules or tablets. Some agents are not available in this formulation and for some medications such as pazopanib, there was a nearly three-fold difference in the MTD for the suspension compared to tablets [136]. Given these pharmacologic variabilities, as well as studies showing that 20% of adults receiving the recommended dose of pazopanib do not achieve therapeutic drug levels [137], some investigators have considered therapeutic drug monitoring [133]. This could prevent long periods of subtherapeutic dosing as well unnecessary overexposure, risk of toxicity and treatment cessation, which are associated with shortened PFS [138]. Monitoring could also resolve current discrepancies with regard to optimal dosing regimens, which may help balance survival and quality of life [137,139]. Widespread implementation of this practice will require a better understanding of the relationship between efficacy and toxicity to exposure for each RTK inhibitor, but provides an additional opportunity to personalize care for the cancer patient.

## 5. Predictive Biomarkers and Clinical Trial Design

The ability to molecularly profile tumors has dramatically increased our understanding of oncogenesis and therapeutic options. Results have now been reported from studies involving several hundred cumulative patients whose tumors were assessed with a variety of testing platforms [6,7,140,141,142]. Common conclusions from these studies include the following: (1) testing from fresh, frozen, or even paraffin-embedded tissue is feasible in a multi-institutional setting; (2) results using comprehensive panels for mutational analysis, gene fusions, and copy number alterations may be available within one month or less; (3) testing identifies specific therapeutic options or suggests changes in therapy in at least half of patients (depending on how one defines an actionable change); and (4) approximately one-fourth of patients with actionable findings are recommended therapy with an RTK inhibitor. Identification of a kinase fusion was the most common reason for recommending a RTK inhibitor, followed by the presence of an activating mutation. Importantly, there have been actionable changes in a wide variety of tumor types, including some that were unexpected and historically not responsive to conventional chemotherapy. 

Outside of treatment for kinase fusions or activating mutations, the absence of similarly strong biomarkers likely dilutes the effectiveness of RTK inhibitors used to treat pediatric solid tumors without these features. It is clear that treatment is beneficial for some sarcoma patients, but we cannot yet reliably identify these patients in a prospective fashion. Combinations of DNA analysis, immunohistochemistry, and proteomic analysis have been proposed, but results are variable and have not been prospectively validated [143,144,145]. Direct measurement of serum proteins such as VEGFA or soluble MET may have some utility in predicting responsiveness to cabozantinib, but requires further confirmation [35]. 

When considering clinical trial design, cooperative group studies are now available which link potentially targetable findings with suitable agents. One large study is the National Cancer Institute-COG Pediatric MATCH study (NCT03155620), which expands the availability of molecular testing and drug availability to over 200 childhood cancer centers. This umbrella trial currently has 13 open arms, including four using RTK inhibitors for patients of any histology with defined targets as assessed by DNA and RNA sequencing: larotrectinib (NTRK), ensartinib (ALK and ROS), erdafitinib (FGFR), and selpercatinib (RET). In the first 1000 patients enrolled on the screening protocol, matches to an open arm were identified in 31% of patients [146]. This rate was higher than expected and may reflect enrollment of patients who had prior molecular testing given the commercial availability of testing platforms. Preliminary results have also been reported from the INFORM study involving eight European countries and over 1300 patients to date. Nearly one-fourth of patients had a very high (8%) or high (14.8%) priority target and patients with very high priority targets treated with matched therapy had improved time to progression [147]. 

Establishing meaningful assessment of clinical benefit will be important in clinical trial design. Traditional endpoints of phase II trials such as objective response rate may be less relevant than progression-free survival in some pediatric cancers [148,149]. For medullary thyroid cancer, assessment of carcinoembryonic antigen or calcitonin may be a surrogate biomarker for response [33]. Early assessment of metabolic response with functional imaging after one month of therapy with cabozantinib identified osteosarcoma and Ewing sarcoma patients with longer progression-free survival [35]. 

Treatment of pediatric solid tumor patients with RTK inhibitors has most often been performed in the setting of single-agent therapy used for recurrent measurable disease. Although this strategy allows for a cleaner assessment of activity, this therapy context may not be optimum for these agents. Given that the genetic landscape of many pediatric solid tumors at diagnosis is relatively simple, an argument has been made for an earlier introduction of these agents before the development of mutations resulting in treatment resistance [150]. Alternatively, maintenance use of RTK inhibitors in the setting of high-risk remission could also be considered given the prior success of using targeted therapy in the treatment of high-risk neuroblastoma in remission [151,152]. 

In regard to dosing, the Innovative Therapies for Children with Cancer consortium released a position paper outlining strategies for early phase clinical trial design for more rapid progress in emerging drug development [153]. Given that targeted therapies tend to have more class-related rather than dose-related toxicities, lengthy dose escalation phases may not be warranted. They recommend that pediatric dosing generally start at adult recommended phase II dose (RP2D) adjusted for body surface area (BSA) provided that this dose should be an equivalent dose of the minimum active target exposure. For drugs with more serious dose-related toxicities, beginning dose escalation at 80% of the adult RP2D is warranted. Due to the wider therapeutic index of many targeted agents, the pediatric RP2D may not need to be as high as the MTD, unless a dose-activity relationship has been documented in adults. Expansion cohorts should be incorporated for populations of patients for whom there is a rational expectation of drug activity. 

## 6. Discussion and Conclusions

The availability of molecular profiling over the past two decades has increased recognition of the potential ways precision medicine can be used for children with solid tumors. The frequency of actionable changes is less in children than adults, and many pediatric malignancies will unfortunately not have molecular changes that can be used to guide therapy. However, there is an important subset of pediatric extracranial solid tumors that do indeed have potentially druggable alterations [8,140,154]. While many of these alterations do not involve RTKs, there remains strong rationale for the use of RTK inhibitors in certain tumor types. For example, testing should be considered for tumors such as infantile fibrosarcoma and inflammatory myofibroblastic tumor that are known to be driven by kinase fusions, given that RTK inhibitors may now be used prior to conventional chemotherapy [60,155]. Similarly, the diagnosis of anaplastic large cell lymphoma or high-risk neuroblastoma often triggers testing for *ALK* mutations that can help direct therapy. Pediatric thyroid cancers also frequently have kinase fusions that are amenable to targeted therapy in situations requiring medical management. In addition, clinicians often pursue testing in other recurrent or metastatic solid tumors in which conventional therapy is unlikely to be curative, with the hope of identifying the rare patient with unexpected actionable findings. However, the fact that many pediatric solid tumors do not have clearly identified predictive biomarkers remains an ongoing challenge for the use of precision medicine in this population. 

Multiple questions about molecular testing are under further investigation. Given that kinase fusions represent the most important biomarker for use of RTK inhibitors, strong consideration should be given for testing that includes some method for their identification, such as RNA sequencing [150]. Other issues being explored include the identification and significance of subclones [156], the optimal timing of sample collection, the significance of tumor heterogeneity, and the ability to screen for biomarkers using circulating tumor cells and cell-free DNA [157,158]. It is clear that our level of testing and identification of more robust predictive biomarkers must improve in order for more patients to benefit from treatment with RTK inhibitors. For example, the combination of a high-throughput small molecule screen complemented with a genome-scale CRISPR-Cas9 gene-knockout screen recently identified several RTKs as therapeutic targets in pediatric rhabdoid tumors [159].

The choice of agent is often based on clinical availability. The median time lag from first-in-human to first-in-child trials for oncology agents is 6.5 years [160], although efforts are underway to hasten access of new drugs to children in the US as well as in developing countries [161,162]. Another access issue is drug formulation, as some younger children may have difficulty with therapy unless an oral solution is available, and even then this may not be tolerable [130]. As our knowledge base grows, we are now able to identify particular mutations which may respond only to second-generation or third-generation agents. Decisions about how RTK inhibitors are best utilized (single-agent or in combination) represent a balance of efficacy and toxicity. While single-agent therapy is very successful in certain contexts [24,33], the combination with chemotherapy will likely be necessary for many RTK inhibitors in order to reduce primary and acquired resistance. These combinations will require close monitoring for toxicity and pharmacokinetic interactions and ultimately randomized trials will be necessary to more clearly assess the benefit of adding RTK inhibitors. Trials adding RTK inhibitors to immunotherapy offer an exciting new avenue for investigation, taking advantage of the effects seen in the tumor microenvironment. Larotrectinib has even been shown to restore the radioactive uptake of iodine in papillary thyroid cancer cells [163], potentially opening up additional lines of investigation.

Clinical trial design will remain a crucial element in optimizing use of these drugs through the identification of appropriate doses and scheduling. The timing of therapy, whether at induction, post-remission, or recurrence, will need to be carefully considered. Management of side effects will also be important given the variable pharmacokinetic profiles of these agents. Although some side effects may correlate with treatment benefit, the cumulative nature of toxicities can negatively affect quality of life. Additional questions remain relative to the appropriate length of therapy, especially for patients with incomplete responses who are tolerating therapy. For patients who go off therapy in remission but subsequently relapse, retreatment with the same RTK inhibitor may be considered although there is little information on this strategy to date [164]. 

In summary, in the 20 years since the first approval of imatinib, great progress has been made in the development of new drugs and the understanding of novel therapeutic targets. However, considerable knowledge gaps remain. It is important for pediatric oncologists to be aware that while the potential of precision medicine is being realized in a few specific pediatric solid tumors, much work remains to extend the applicability of RTK inhibitors through biomarker identification and thoughtful clinical testing.

## Figures and Tables

**Table 1 cancers-13-03531-t001:** Select completed trials of receptor tyrosine kinase inhibitors for pediatric solid tumors ^†^.

Agent	Key Targets Inhibited	Relevant Tumors	Key Toxicities (Grade 3–4)	Comments
Larotrectinib [24,25]	TRKA, TRB, TRKC (IC_50_ 5–11 nM)	Infantile fibrosarcoma, salivary gland tumor, mesoblastic nephroma, lymphoma, solid tumors	Transaminitis, anemia, neutropenia	75–92% ORR; USFDA approved for pediatric solid tumors harboring *NTRK* gene fusions
Crizotinib [26,27,28,29]	ALK (IC_50_ 24 nM) c-MET (IC_50_ 5–20 nM)	ALCL, IMT, NBL	Neutropenia, diarrhea	>80% ORR for ALCL or IMT; 9% ORR for NBL; USFDA approved for relapsed ALK+ ALCL ages 1–21 years
Entrectinib [30,31,32]	ALK (IC_50_ 12 nM) TRKA-C (IC_50_ 1–5 nM) ROS1 (IC_50_ 7 nM)	NBL, salivary gland tumor, sarcoma, thyroid cancer	Fatigue, weight gain, transaminitis, myelosuppression, hyperuricemia	57% ORR; USFDA approved for patients age ≥ 12 years with metastatic/unresectable solid tumors harboring *NTRK* gene fusions
Selpercatinib [33,34]	RET (IC_50_ 4 ± 2 nM)	Thyroid cancer, malignant peripheral nerve sheath tumors, sarcomas	Hypertension, diarrhea, transaminitis, prolonged QT interval on electrocardiogram	69–79% ORR; 64–92% PFS at 1 year; USFDA approved for RET-mutant thyroid cancers in patients age ≥12 years
Cabozantinib [35,36]	VEGFR2 (IC_50_ 0.035 nM) c-MET (IC_50_ 1.3 nM) KIT (IC_50_ 4.6 nM) RET (IC_50_ 5.2 nM)	Ewing sarcoma, osteosarcoma	Hypophosphatemia, transaminitis, HFSR, pneumothorax, neutropenia	Osteosarcoma: 17% ORR, with PFS 52% at 6 months Ewing sarcoma: 26% ORR; with PFS 33% at 6 months
Regorafenib [37,38,39,40,41]	VEGFR1/2 (IC_50_ 4.2-13 nM) KIT (IC_50_ 7 nM) RET (IC_50_ 1.5 nM) PDGFRβ (IC_50_ 22 nM)	Ewing sarcoma, osteosarcoma	Fatigue, chest pain, hypophosphatemia, HFSR, hypertension, alkaline phosphatase, myelosuppression, diarrhea, mucositis, hypertension	Osteosarcoma: ORR 8-14%, with 44–62% PFS at 4 months Ewing sarcoma: ORR 10–22%; 56% PFS at 8 weeks, 26% PFS at 6 months
Lenvatinib [42,43]	VEGFR2 (IC_50_ 4 nM) RET (IC_50_ 6.4 nM) PDGFRα (IC_50_ 29 nM) FGFR2 (IC_50_ 27 nM)	Osteosarcoma	Back pain, dyspnea	ORR 7%, with 33.3% PFS at 4 months
Sorafenib [44,45,46]	VEGFR2 (IC_50_ 4 nM) RET (IC_50_ 0.4 nM) PDGFRα (IC_50_ 18 nM)	Osteosarcoma	HFSR, thrombocytopenia, anemia, creatine kinase elevation	14% ORR; 46% PFS at 4 months; Six-month PFS increased from 29% to 45% with the addition of everolimus
Apatinib [47,48]	VEGFR2 (IC_50_ 1 nM) RET (IC_50_ 13 nM)	Osteosarcoma	Pneumothorax, wound dehiscence, proteinuria, diarrhea HFSR	ORR 43%, with 57% PFS at 4 months;
Pazopanib [49,50,51]	VEGFR1 (IC_50_ 10 nM)	NRSTS given in combination with ifosfamide and doxorubicin	Myelosuppression, febrile neutropenia, sepsis, emesis, wound dehiscence	58% of patients treated with pazopanib had pathological response ≥90% vs. 22% with chemo alone

Legend: IC_50_ = half maximal inhibitory concentration, nM = nanomolar, TRK = tropomyosin receptor kinase, ALK = anaplastic lymphoma kinase, c-MET = hepatocyte growth factor receptor, RET = rearranged during transfection, VEGFR = vascular endothelial growth factor receptor, PDGFR = platelet-derived growth factor receptor, FGFR = fibroblast growth factor receptor, ORR = overall response rate, PFS = progression-free survival, USFDA = United States Food and Drug Administration, ALCL = anaplastic large cell lymphoma, IMT = inflammatory myofibroblastic tumor, NBL = neuroblastoma, NRSTS = non-rhabdomyosarcoma soft-tissue sarcoma, HFSR = hand–foot skin reaction, ^†^ = due to trial design these results include a significant number of adult patients.

**Table 2 cancers-13-03531-t002:** Select ongoing clinical trials of receptor tyrosine kinase inhibitors in pediatric solid tumors.

Agent	Disease	Study Population and Additional Details	ClinicalTrials.gov Identifier
Crizotinib	*ALK*-altered neuroblastoma	Newly diagnosed, high-risk patients; given with standard therapy	NCT03126916
Ensartinib	*ALK-* or *ROS1*-altered solid tumors, histiocytic disorders	Recurrent/refractory advanced disease; investigating biomarkers	NCT03213652
Entrectinib	*NTRK1/2/3* or *ROS1* fusion-positive solid tumors	Recurrent/refractory disease	NCT02650401
Erdafitinib	*FGFR*-mutated solid tumors, NHL, histiocytic disorders	Recurrent/refractory advanced disease	NCT03210714
Larotrectinib	*NTRK* fusion-positive solid tumors, NHL, histiocytic disorders	Recurrent or refractory advanced disease	NCT03213704
Lenvatinib	Phase I: all solid tumors Phase II: Ewing sarcoma, rhabdomyosarcoma	Recurrent/refractory disease; given with everolimus	NCT03245151
Lorlatinib	*ALK*-altered neuroblastoma	Phase I, alone or in combination with conventional chemotherapy	NCT03107988
Regorafenib	Multiple bone and soft tissue sarcoma types	Recurrent/refractory advanced disease	NCT02048371
Regorafenib	Osteosarcoma	Recurrent/refractory disease; given with nivolumab	NCT04803877
Repotrectinib	Solid tumors with *ALK*, *ROS1*, or *NTRK1/2/3* alterations	Recurrent/refratcory disease	NCT04094610
Selpercatinib	*RET*-altered solid tumors, lymphomas, histiocytic disorders	Recurrent/refractory advanced disease	NCT04320888
Sorafenib	Hepatocellular carcinoma	Newly diagnosed advanced disease; given with chemotherapy	NCT03017326
Ceritinib	ALK-altered neuroblastoma	Recurrent/refractory disease; given with ribociclib	NCT02780128

Abbreviations: HSCT = hematopoietic stem cell transplant, NHL = Non-Hodgkin lymphoma, DIPG = diffuse intrinsic pontine glioma, CNS = central nervous system, PK = pharmacokinetic.

## Data Availability

No new data were created or analyzed in this study. Data sharing is not applicable to this article.

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
