# Peer review of "Pursuing Precision: Receptor Tyrosine Kinase Inhibitors for Treatment of Pediatric Solid Tumors"

_cancers, 2021, doi:10.3390/cancers13143531_

Round 1
Reviewer 1 Report
In this paper entitled "Pursuing Precision: Receptor Tyrosine Kinase Inhibitors for Treatment of Pediatric Solid Tumors" the authors review the use of RTK inhibitors in patients with different types of pediatric cancer, including the results of clinical trials that have already been completed and published and mentioning other trials that are currently underway. The review is well written and easy to follow. In addition, it gives visibility to the problem encountered by pediatric oncologists: in many cases it is not possible to extrapolate what has been observed with treatments in adult tumors to what happens in children, and it is not easy to carry out clinical trials due to the low number of patients.
Although the review is interesting, I propose some details that could further improve the quality of the article:
1. It is true that precision medicine in both adults and children is a goal of current medicine and that "genetic testing platforms has improved our understanding of the specific molecular signatures", as stated in the text. However, it has been shown that only half of pediatric tumors have an identifiable therapeutic target (e.g. Gröbner, S., Worst, B., Weischenfeldt, J. et al. The landscape of genomic alterations across childhood cancers. Nature 555, 321-327 (2018); Campbell BB, Light N, Fabrizio D, et al. Comprehensive Analysis of Hypermutation in Human Cancer. Cell. 2017 Nov 16;171(5):1042-1056.e10.). This should be mentioned in the text to recognize that genetic tests are not helpful in many cases and that precision medicine is an even greater challenge in pediatric tumors.- Table 1 is a bit chaotic to see what are the targets of each inhibitor and in what type of tumors it is being used. Perhaps it would make it easier to separate the results by tumor type or even make a figure indicating what the targets of each inhibitor are. In addition, there are some trials that include pediatric and adult patients and do not discriminate the results between patients. If the inhibitor has been approved for any age, as in the case of larotretinib, it is understood that this is because it has worked well in children, but it is important to know that the trial is not exclusively pediatric. Furthermore, in cases that include children and adults and do not separate the results of both, it could be interesting to include a published pediatric clinical case that confirms the good results described in the trial.
- Angiogenesis is the main mechanism by which blood vessels are formed in tumors, but there are others such as co-option, vasculogenic mimicry, vessel intussusception or vasculogenesis. The adjective "main" is very relevant in this context. It should also be noted that when the text says that VEGFR2 can be inhibited in the absence of mutations, it is because this receptor is normally overexpressed in endothelial cells and not in tumor cells, so that when angiogenesis inhibitors are used, the therapy is directed against abiological process necessary for tumor growth and not against the genetic alterations that transform a normal cell into a tumor cell. In my opinion this is not clear in the text and it is important when defending that the use of RTK brings the therapy closer to personalized medicine.
- The relationship of VEGF to the immunosuppressive microenvironment is much more complex than saying that this factor creates a microenvironment that recruits MDSCs and Tregs. In fact, although it is a field in which much remains to be discovered, it is likely that the reason for the increase in VEGF and the recruitment of these cell types is hypoxia and the cellular and non-cellular composition of the tumor, so that the statement "VEGF can create an immunosuppressive environment through the recruitment of myeloid-derived suppressor cells and regulatory T-cells .." (lines 304-305) is too simplistic and not entirely correct.
- At present some of the pathways mentioned in this review are inhibited by blocking the binding of ligands to these receptors. To give an example, the most important and most widely used inhibitor of the VEGF pathway is Bevacizumab. It is true that it is not the objective of the review, but it is important to cite it in the discussion.
Minor comments:
1. VEGF is Vascular ENDOTHELIAL growth factor and not epidermal (line 64).
2. In table 1, Lenvatinib also inhibits FGFR, Sorafenib PDGFR and pazopanib PDGFR, FGFR. This should be included in the table.
3. In table 1, in the key targets inhibited of pazopanib change combinaiton to combination.
4. In line 123, in vitro and in vivo results are linked with the conjunction "with", but since they are independent experiments, it is better to join them with "and".
5. Several clinical trials registered in Clinical trials are mentioned throughout the manuscript and then listed in Table 2. This table should be mentioned in the text where these trials appear. In addition, trials NCT03155620 and NCT01956669 appear in the text but are not included in the table.
6. In the last paragraph of section 2.1. the FGFR receptor is mentioned, but this is one of the most relevant signaling pathways of angiogenesis. I recommend changing this paragraph to that section.
Despite these comments, I congratulate the authors on their work.
Author Response
Reviewer: 1
Comments to the Author
In this paper entitled "Pursuing Precision: Receptor Tyrosine Kinase Inhibitors for Treatment of Pediatric Solid Tumors" the authors review the use of RTK inhibitors in patients with different types of pediatric cancer, including the results of clinical trials that have already been completed and published and mentioning other trials that are currently underway. The review is well written and easy to follow. In addition, it gives visibility to the problem encountered by pediatric oncologists: in many cases it is not possible to extrapolate what has been observed with treatments in adult tumors to what happens in children, and it is not easy to carry out clinical trials due to the low number of patients.
Although the review is interesting, I propose some details that could further improve the quality of the article:
- It is true that precision medicine in both adults and children is a goal of current medicine and that "genetic testing platforms has improved our understanding of the specific molecular signatures", as stated in the text. However, it has been shown that only half of pediatric tumors have an identifiable therapeutic target (e.g. Gröbner, S., Worst, B., Weischenfeldt, J. et al. The landscape of genomic alterations across childhood cancers. Nature 555, 321-327 (2018); Campbell BB, Light N, Fabrizio D, et al. Comprehensive Analysis of Hypermutation in Human Cancer. Cell. 2017 Nov 16;171(5):1042-1056.e10.). This should be mentioned in the text to recognize that genetic tests are not helpful in many cases and that precision medicine is an even greater challenge in pediatric tumors.
The authors thank the reviewer for their insights, and agree with their assessment of the challenges faced by pediatric oncologists. Accordingly, we have modified the first paragraph of the discussion as shown below.
“The availability of molecular profiling over the past two decades has increased recognition of the potential ways precision medicine can be used for children with solid tumors. The frequency of actionable changes is less in children than adults, and even with comprehensive testing many patients will unfortunately not have molecular changes that can be used to guide therapy. However, there is an important subset of pediatric extracranial solid tumors that do indeed have potentially druggable alterations. While many of these alterations do not involve RTKs, there remains strong rationale for the use of RTK inhibitors in certain tumor types. For example, testing should be considered for tumors known to be driven by kinase fusions such as infantile fibrosarcoma and inflammatory myofibroblastic tumors, given that RTK inhibitors may now be used prior to conventional chemotherapy. Similarly, the diagnosis of anaplastic large cell lymphoma or high-risk neuroblastoma often triggers testing for ALK mutations that can help direct therapy. Pediatric thyroid cancers also frequently have kinase fusions that are amenable to targeted therapy in situations requiring medical management. In addition, clinicians often pursue testing in other recurrent or metastatic solid tumors in which conventional therapy is unlikely to be curative, with the hope of identifying the rare patient with unexpected actionable findings. The fact that many pediatric solid tumors do not have clearly identified predictive biomarkers for use of targeted agents has made precision medicine an ongoing challenge for this population.”
Gröbner, S.N.; Worst, B.C.; Weischenfeldt, J.; Buchhalter, I.; Kleinheinz, K.; Rudneva, V.A.; Johann, P.D.; Balasubramanian, G.P.; Segura-Wang, M.; Brabetz, S.; et al. The landscape of genomic alterations across childhood cancers. Nature 2018, 555, 321-327, doi:10.1038/nature25480.
Harris, M.H.; DuBois, S.G.; Glade Bender, J.L.; Kim, A.; Crompton, B.D.; Parker, E.; Dumont, I.P.; Hong, A.L.; Guo, D.; Church, A.; et al. Multicenter Feasibility Study of Tumor Molecular Profiling to Inform Therapeutic Decisions in Advanced Pediatric Solid Tumors: The Individualized Cancer Therapy (iCat) Study. JAMA oncology 2016, 2, 608-615, doi:10.1001/jamaoncol.2015.5689.
Campbell, B.B.; Light, N.; Fabrizio, D.; Zatzman, M.; Fuligni, F.; de Borja, R.; Davidson, S.; Edwards, M.; Elvin, J.A.; Hodel, K.P.; et al. Comprehensive Analysis of Hypermutation in Human Cancer. Cell 2017, 171, 1042-1056.e1010, doi:10.1016/j.cell.2017.09.048.
- Table 1 is a bit chaotic to see what are the targets of each inhibitor and in what type of tumors it is being used. Perhaps it would make it easier to separate the results by tumor type or even make a figure indicating what the targets of each inhibitor are. In addition, there are some trials that include pediatric and adult patients and do not discriminate the results between patients. If the inhibitor has been approved for any age, as in the case of larotretinib, it is understood that this is because it has worked well in children, but it is important to know that the trial is not exclusively pediatric. Furthermore, in cases that include children and adults and do not separate the results of both, it could be interesting to include a published pediatric clinical case that confirms the good results described in the trial.
The authors appreciate the thoughtful suggestions of the reviewer and their careful evaluation of this table. We considered various approaches to presenting this information, but found that using other formats only made the table more challenging to read. After lengthy reflection, we concluded that the current structure of the table offers the reader key information on the drugs tested, the main targets of those drugs, the most relevant tumor types studied, how therapy was tolerated, and what were the most important findings. We have added a designation indicating the inclusion of adult patients in the reported data. We are happy to make further changes in this table if the editor feels that will facilitate readability.
- Angiogenesis is the main mechanism by which blood vessels are formed in tumors, but there are others such as co-option, vasculogenic mimicry, vessel intussusception or vasculogenesis. The adjective "main" is very relevant in this context. It should also be noted that when the text says that VEGFR2 can be inhibited in the absence of mutations, it is because this receptor is normally overexpressed in endothelial cells and not in tumor cells, so that when angiogenesis inhibitors are used, the therapy is directed against abiological process necessary for tumor growth and not against the genetic alterations that transform a normal cell into a tumor cell. In my opinion this is not clear in the text and it is important when defending that the use of RTK brings the therapy closer to personalized medicine.
To address this point, we have reworded the relevant paragraph to now state, “Even when an activating mutation or gene fusion is not identified in tumor cells, RTK inhibitors still may be useful, although durable benefit is less likely. This strategy has primarily focused on targeting angiogenesis, which is the main mechanism for the new blood vessel development that is essential for tumor growth and metastasis. Although this process is complex and driven by many factors, signaling through the vascular endothelial growth factor receptor 2 (VEGFR2) is particularly important. VEGFR2 is often overexpressed by the endothelial cells of solid tumors, and can be inhibited at low nanomolar levels by several of the RTK inhibitors even in the absence of a defined gene mutation or fusion. In this way, the RTK inhibitor is acting to disrupt the supply of nutrients to the tumor rather than directly inhibit a specific oncologic driver.”
- The relationship of VEGF to the immunosuppressive microenvironment is much more complex than saying that this factor creates a microenvironment that recruits MDSCs and Tregs. In fact, although it is a field in which much remains to be discovered, it is likely that the reason for the increase in VEGF and the recruitment of these cell types is hypoxia and the cellular and non-cellular composition of the tumor, so that the statement "VEGF can create an immunosuppressive environment through the recruitment of myeloid-derived suppressor cells and regulatory T-cells .." (lines 304-305) is too simplistic and not entirely correct.
The reviewer astutely highlights the complexity of the underlying physiology of the tumor microenvironment. Accordingly, we have revised the paragraphs of this sections as shown below
“Aberrant angiogenesis can promote a hostile tumor microenvironment that is hypoxic and acidotic. RTKs have also been shown to modulate non-endothelial components of the tumor microenvironment through complex mechanisms such as enhancement of antitumor immunity and local immune cell populations. One example is the role of VEGF in creating an immunosuppressive environment through the recruitment of myeloid-derived suppressor cells and regulatory T-cells, as well as the inhibition of dendritic cell maturation and antigen presentation. A deeper understanding of tumor microenvironment and its role in cancer is needed to aid in the design of future clinical trials.
To begin to address these complicated changes, Wilky et al. combined the VEGFR inhibitor axitinib with the anti-programmed death 1 (PD-1) antibody pembrolizumab to treat adults with sarcoma. The 3-month PFS of 65% and response rate of 25% exceeded historical expectations. Similarly encouraging preliminary results were also seen with a trial of sunitinib and the anti-PD-1 antibody nivolumab for adult bone sarcoma. Additional studies are underway to further assess the ability of RTK inhibitors to synergize with immunotherapies, such as a trial of regorafenib and nivolumab for patients with recurrent osteosarcoma (NCT04803877). However, there is a potential that combination therapy may exacerbate toxicity, as seen in a study combining crizotinib and nivolumab in lung cancer patients which was terminated early due to severe hepatic toxicity.”
- At present some of the pathways mentioned in this review are inhibited by blocking the binding of ligands to these receptors. To give an example, the most important and most widely used inhibitor of the VEGF pathway is Bevacizumab. It is true that it is not the objective of the review, but it is important to cite it in the discussion.
The reviewer raises an important issue, and we have added the following statements starting in line 313:
“Another method for targeting angiogenesis in solid tumors is the use of bevacizumab, a monoclonal antibody that binds to VEGF and interferes with binding to VEGFRs. This strategy has a different mechanism of inhibition that is narrower than seen with some RTK inhibitors. Previous studies have shown limited activity of bevacizumab as a single agent, and in combination with chemotherapy for pediatric sarcoma. However, no direct comparisons of anti-VEGF antibodies versus RTK inhibitors in pediatric solid tumor patients have been reported.”
Glade Bender, J.L.; Adamson, P.C.; Reid, J.M.; Xu, L.; Baruchel, S.; Shaked, Y.; Kerbel, R.S.; Cooney-Qualter, E.M.; Stempak, D.; Chen, H.X.; et al. Phase I trial and pharmacokinetic study of bevacizumab in pediatric patients with refractory solid tumors: a Children's Oncology Group Study. Journal of clinical oncology : official journal of the American Society of Clinical Oncology 2008, 26, 399-405, doi:10.1200/jco.2007.11.9230.
Chisholm, J.C.; Merks, J.H.M.; Casanova, M.; Bisogno, G.; Orbach, D.; Gentet, J.C.; Thomassin-Defachelles, A.S.; Chastagner, P.; Lowis, S.; Ronghe, M.; et al. Open-label, multicentre, randomised, phase II study of the EpSSG and the ITCC evaluating the addition of bevacizumab to chemotherapy in childhood and adolescent patients with metastatic soft tissue sarcoma (the BERNIE study). European journal of cancer (Oxford, England : 1990) 2017, 83, 177-184, doi:10.1016/j.ejca.2017.06.015.
Navid, F.; Santana, V.M.; Neel, M.; McCarville, M.B.; Shulkin, B.L.; Wu, J.; Billups, C.A.; Mao, S.; Daryani, V.M.; Stewart, C.F.; et al. A phase II trial evaluating the feasibility of adding bevacizumab to standard osteosarcoma therapy. International journal of cancer 2017, 141, 1469-1477, doi:10.1002/ijc.30841.
Minor comments:
1. VEGF is Vascular ENDOTHELIAL growth factor and not epidermal (line 64).
Thank you for identifying this error. We have addressed it in the manuscript.
In table 1, Lenvatinib also inhibits FGFR, Sorafenib PDGFR and pazopanib PDGFR, FGFR. This should be included in the table.
We have added to Table 1 the IC50 values describing inhibition of FGFR2 by lenvatinib, and inhibition of PDGFRα by sorafenib. The extent of inhibition of PDGFR and FGFR by pazopanib (IC50 of 84 and 140 nM, respectively) is less robust than other targets included in this table, and so were not listed.
Harris PA, Boloor A, Cheung M, et al. Discovery of 5-[[4-[(2,3-dimethyl-2H-indazol-6-yl)methylamino]-2-pyrimidinyl]amino]-2-methyl-benzenesulfonamide (Pazopanib), a novel and potent vascular endothelial growth factor receptor inhibitor. J Med Chem. 2008;51(15):4632-4640. doi:10.1021/jm800566m
In table 1, in the key targets inhibited of pazopanib change combinaiton to combination.
The authors appreciate the reviewer’s identification of this typo, and have made the necessary correction.
In line 123, in vitroand in vivoresults are linked with the conjunction "with", but since they are independent experiments, it is better to join them with "and".
The reviewer’s suggestion provides clarity for the reader and we have adjusted the wording in the manuscript.
Several clinical trials registered in Clinical trials are mentioned throughout the manuscript and then listed in Table 2. This table should be mentioned in the text where these trials appear. In addition, trials NCT03155620 and NCT01956669 appear in the text but are not included in the table.
The clinical trial NCT03155620 refers to the umbrella Pediatric MATCH trial, individual arms of which utilize RTK inhibitors have been included in the table. Line 221 has been updated to reflect the subprotocol (NCT03210714) for FGFR-altered solid tumors. NCT01956669 on the other hand is a completed trial, and therefore not included in this table. Table 2 has been referenced in the body of the text as recommended.
In the last paragraph of section 2.1. the FGFR receptor is mentioned, but this is one of the most relevant signaling pathways of angiogenesis. I recommend changing this paragraph to that section.
The authors appreciate this final comment regarding the role of FGFR in angiogenesis. This paragraph focused on instances of FGFR mutation or overexpression, which typically provides an opportunity for more effective therapy than when angiogenesis is targeted alone. However it is positioned as such to provide a natural transition to the section on angiogenesis, and the paragraph was revised to state:
“Fibroblast growth factor receptor (FGFR) is a family of RTKs with downstream targets that promote cell proliferation, survival, and migration, including MAPK and PI3K. FGFR is often overexpressed or mutated in rhabdomyosarcoma. Erdafitinib is a pan-FGFR inhibitor with activity at levels as low as 1 nanomolar, and is approved for the treatment of metastatic urothelial cancer in adults. A prospective phase II trial for pediatric solid tumors with FGFR alterations is now being conducted (NCT03210714, Table 2). FGFR signaling also plays a prominent role in angiogenesis, which provides an alternative avenue for the use of RTK inhibitors.”
Despite these comments, I congratulate the authors on their work.
Reviewer 2 Report
This is a fine paper and a wonderful narrative around the use of Receptor Tyrosine Kinases in pediatric cancers. I have on issues with this review - which is well structured comprehensive, sufficiently critical and up-to-date with the latest state of the field. Well done. Happy to support it immediate publication.
Author Response
Comments to the Author
This is a fine paper and a wonderful narrative around the use of Receptor Tyrosine Kinases in pediatric cancers. I have on issues with this review - which is well structured comprehensive, sufficiently critical and up-to-date with the latest state of the field. Well done. Happy to support it immediate publication.
The authors appreciate the time Reviewer #2 took to ensure the quality of this manuscript, and thank them for their feedback.
Reviewer 3 Report
The review states a good source of information about new targeted therapies in treatment of pediatric solid tumors.
I have some questions: - how many patients with particular solid tumors have specific genes fusions?, - did the Authors meet the trials which combine larotrectinib with chemotherapy?, - when is CR what is recommended time of therapy for each agent? these data are very important, -do the Authors have information about targeted therapies in Wilms tumors or pediatric GCTs?, I have seen clinical trials only with sarcomas, mesoblastic nephromas, salivary gland tumors, thyroid cancers, Ewing sarcomas, osteosarcomas, MPNSTs, hepatocellular carcinomas (the tables 1 and 2).
Author Response
Comments to the Author
The review states a good source of information about new targeted therapies in treatment of pediatric solid tumors. I have some questions:
- how many patients with particular solid tumors have specific genes fusions?
Molecular changes in pediatric solid tumors range in scale from single nucleotide variants to larger structural changes. As we mention regarding genomic testing in our manuscript:
“Common conclusions from these studies include: 1) testing from fresh, frozen, or even paraffin-embedded tissue is feasible in a multi-institutional setting; 2) results using comprehensive panels for mutational analysis, gene fusions, and copy number alterations may be available within one month or less; 3) testing identifies specific therapeutic options or suggests changes in therapy in at least half of patients (depending on how one defines an actionable change); and 4) approximately one-fourth of patients with actionable findings are recommended therapy with an RTK inhibitor. Identification of a kinase fusion was the most common reason for recommending a RTK inhibitor, followed by the presence of an activating mutation. Importantly, there have been actionable changes in a wide variety of tumor types, including some that were unexpected and historically not responsive to conventional chemotherapy.”
- did the Authors meet the trials which combine larotrectinib with chemotherapy?
The authors are unaware of any major pediatric trials which combine larotrectinib with chemotherapy. We would be pleased to include any relevant trials which utilize combination of larotrectinib with chemotherapy for pediatric solid tumors that the reviewer suggests.
- when is CR what is recommended time of therapy for each agent? these data are very important
We have previously included the time to response with larotrectinib (“The median time to response was at the first 8-week scheduled assessment”). We have added information about time to response in patients with anaplastic large cell lymphoma or inflammatory myofibroblastic tumor treated with crizotinib (“The median time to response was within the first month of therapy”). Responses for patients with sarcoma or other solid tumors are infrequent and the time to response is often not reported.
- do the Authors have information about targeted therapies in Wilms tumors or pediatric GCTs?, I have seen clinical trials only with sarcomas, mesoblastic nephromas, salivary gland tumors, thyroid cancers, Ewing sarcomas, osteosarcomas, MPNSTs, hepatocellular carcinomas (the tables 1 and 2).
Unfortunately, only limited information is available regarding the use of RTK inhibitors for treatment of Wilms tumor. As mentioned in the manuscript (lines 286-293), an exceptional response was seen in one Wilms tumor patient treated with cabozantinib, but no responses were seen in larger multi-institutional phase II trials of cabozantinib or sorafenib. We are not aware of additional published information regarding RTK inhibitors in this population.
Similarly, the experience of RTK inhibitors for treating recurrent germ cell tumors has also been limited. We have added the following information about this tumor type, as well as desmoid tumors: “The use of RTK inhibitors has also been investigated for treatment of other uncommon pediatric solid tumors. Although germ cell tumors have been reported to express c-KIT and PDGFR, inhibitors such as sunitinib, pazopanib, and sorafenib have shown only limited activity. In contrast, sorafenib appeared beneficial for treatment of adults with desmoid tumors, with a response rate of 33% and doubling of the 2-year PFS compared to placebo. The use of sorafenib or pazopanib is now being explored for younger patients as well.
Galvez-Carvajal, L.; Sanchez-Muñoz, A.; Ribelles, N.; Saez, M.; Baena, J.; Ruiz, S.; Ithurbisquy, C.; Alba, E. Targeted treatment approaches in refractory germ cell tumors. Critical reviews in oncology/hematology 2019, 143, 130-138, doi:10.1016/j.critrevonc.2019.09.005.
Gounder, M.M.; Mahoney, M.R.; Van Tine, B.A.; Ravi, V.; Attia, S.; Deshpande, H.A.; Gupta, A.A.; Milhem, M.M.; Conry, R.M.; Movva, S.; et al. Sorafenib for Advanced and Refractory Desmoid Tumors. The New England journal of medicine 2018, 379, 2417-2428, doi:10.1056/NEJMoa1805052.
Agresta, L.; Kim, H.; Turpin, B.K.; Nagarajan, R.; Plemmons, A.; Szabo, S.; Dasgupta, R.; Sorger, J.I.; Pressey, J.G. Pazopanib therapy for desmoid tumors in adolescent and young adult patients. Pediatric blood & cancer 2018, 65, e26968, doi:10.1002/pbc.26968.
Robles, J.; Keskinyan, V.S.; Thompson, M.; Davis, J.T.; Mater, D.V. Combination therapy with sorafenib and celecoxib for pediatric patients with desmoid tumor. Pediatric hematology and oncology 2020, 37, 445-449, doi:10.1080/08880018.2020.1735591.